



# Long-term studies of MLT summer length definitions based on mean zonal wind features observed for more than one solar cycle at mid- and high-latitudes in the northern hemisphere

Juliana Jaen[1], Toralf Renkwitz[1], Jorge L. Chau[1], Maosheng He[1], Peter Hoffmann[1,†], Yosuke Yamazaki[2], Christoph Jacobi[3], Masaki Tsutsumi[4], Vivien Matthias[5], and Chris Hall[6,††]

[1]Leibniz-Institute of Atmospheric Physics at the University of Rostock, Schloss-Strasse 6, 18225 Kühlungsborn, Germany
[2]GFZ German Research Centre for Geosciences, Potsdam, Germany
[3]Institute Meteorology, Leipzig University, Stephanstr. 3, 04103 Leipzig, Germany
[4]National Institute of Polar Research, Tokyo, Japan
[5]Institute for Solar-Terrestrial Physics, German Aerospace Center (DLR), Neustrelitz, Germany
[6]Tromsø Geophysical Observatory, The Arctic University of Norway, Tromsø, Norway
[†]deceased, 29 October 2020
[††]deceased, 9 August 2021

**Correspondence:** Juliana Jaen (jaen@iap-kborn.de)

**Abstract.**

Specular meteor radars (SMRs) and partial reflection radars (PRRs) have been observing mesospheric winds for more than a solar cycle over Germany (∼54°N) and northern Norway (∼69°N). This work investigates the mesospheric mean zonal wind and the zonal mean geostrophic zonal wind from the Microwave Limb Sounder (MLS) over these two regions between 2004 and 2020. Our study focuses on the summer when strong planetary waves are absent and the stratospheric and tropospheric conditions are relatively stable. We establish two definitions of the summer length according to the zonal wind reversals: (1) the mesosphere and lower thermosphere summer length (MLT-SL) using SMR and PRR winds, and (2) the mesosphere summer length (M-SL) using PRR and MLS. Under both definitions, the summer begins around April and ends around mid-September. The largest year to year variability is found in the summer beginning in both definitions, particularly at high-latitudes, possibly due to the influence of the polar vortex. At high-latitudes, the year 2004 has a longer summer length compared to the mean value for MLT-SL, as well as 2012 for both definitions. The M-SL exhibits an increasing trend over the years, while MLT-SL does not have a well-defined trend. We explore a possible influence of solar activity, as well as large-scale atmospheric influences (e.g. quasi-biennial oscillations (QBO), El Niño-southern oscillation (ENSO), major sudden stratospheric warming events). We complement our work with an extended time series of 31 years at mid-latitudes using only PRR winds. In this case, the summer length shows a breakpoint, suggesting a non-uniform trend, and periods similar to those known for ENSO and QBO.

*Copyright statement.* TEXT



## 1 Introduction

As Earth orbits around the Sun, the duration of four seasons is well defined at ground level at mid-latitudes. Higher up between
50 and 100 km in the mesosphere and lower thermosphere (MLT) the separation is not well defined. The Earth's atmosphere
is a complex system governed by several processes that are continuously evolving (e.g. radiative heating, coupling, mixing
processes, etc.). The dynamics of the MLT are forced mainly by solar radiation, and the wave activities arising in the lower
atmosphere, such as planetary waves, gravity waves, tides (e.g. Yiğit et al., 2016). Circulation patterns such as the stratospheric
quasi-biennial oscillation (QBO, Baldwin et al., 2001) and El Niño-Southern Oscillation (ENSO, Wang and Picaut, 2004) also
influence the MLT dynamics at midlatitudes. During winter conditions strong planetary wave activity is present and later on, in
the transition from winter to summer, a reduction of the planetary activity occurs (Lauter and Entzian, 1983; Hoffmann et al.,
2002). With the transition, every year the zonal wind circulation in the MLT displays the final reversal of the wind direction
from eastward to westward, in part produced by the wave dissipation generated by gravity wave activity (see Hoffmann et al.,
2010; Laskar et al., 2017). In connection to this wind reversal, the mesopause experiences a decrease in temperature resulting
in the appearance of ice particles, due to the water vapor present in the atmosphere. The presence of charged ice particles is
observed through radar echoes known as Polar Mesospheric Summer Echoes (PMSE, see e.g. Rapp et al., 2003). Between 80
and 90 km and on nanometer scales, a congregation of ice particles is called noctilucent clouds or polar mesospheric clouds
(e.g. Hervig et al., 2001; Baumgarten et al., 2008; Fiedler et al., 2015).

The above-mentioned summer characteristics exhibit interannual variabilities and interactions with adjacent layers, high-
lighting the importance of studying this season as well as its long-term behavior. Previous works, like Lauter and Entzian
(1983) used the mean zonal wind reversal at around 25 km altitude to study long-term behavior between 1958 and 1982, ob-
taining an increase in the summer duration of 0.52 days per year (d/yr). Nevertheless, the authors suggested a possible change
in the trends after 1980 and reported a connection between the QBO and the wind reversal dates during 1958-1967 and again
after 1977. Under the name of *Equivalent Summer Duration*, defined by a threshold of 198 K at 87 km at mid-latitudes, the
temperature variations during summer were investigated by Offermann et al. (2010). This study covered the years 1988 to
2008, obtaining a change rate of 1.21 d/yr. The authors also compared the obtained values with the zonal wind reversal in the
stratosphere (20 hPa) that showed a decrease in the summer duration of 0.99 d/yr. Offermann et al. (2005) characterized the
summer duration referring to the stratospheric zonal wind reversal in broad ranges of latitude and altitude between 1948 and
2003. They found a dependency in latitude and altitude, with longer summer duration at higher altitudes and high-latitudes
for the northern hemisphere. However, in all the cases they detected a breakpoint around 1978-1980, obtaining an increase of
summer duration before the breakpoint and negative trends after 1978/1980.

For several years, long-term studies aimed to investigate the anthropogenic influence in the atmosphere. Having in mind the
atmosphere as a whole and considering that the temperature changes affect life at ground level, we can compare the summer
length with the vegetation growing season. The normalized difference vegetation index (NDVI) is obtained from $CO_2$ satellite
measurements (e.g. Zhou et al., 2001). During the vegetation growing season there is an exchange of $CO_2$ between the plants
and the soil due to the process of photosynthesis and decomposition (Fung et al., 1987). In these studies the summer season





starts around April/May and ends in September/October at mid-latitudes, that appears to be comparable to the time-interval for the mean zonal wind reversal investigated in this manuscript.

The middle atmosphere studies, mentioned above, were made before 2010. Since the beginning of the 21st century, new radar
systems were deployed in Germany and Norway. After more than one solar cycle of system operations we are able to look into the long-term trends. In this study, we aim to analyze the long-term variability of the mean zonal wind reversal (MZWR), which occurs around March and in September by implementing two different definitions of summer length. Both definitions applied to radar wind measurements are related to different processes and regions in MLT as well as incorporate altitude and latitude features. With these definitions we search for possible correlations with known forcing events from above or below the MLT,
like e.g. solar activity measured by Lyman$-\alpha$ line, major sudden stratospheric warming (MSSW, Butler et al., 2015), strong polar-night jet oscillations (sPJO, Peters et al., 2018; Conte et al., 2019). The Lyman $\alpha$ line is a good representation of the solar activity in the MLT, as a consequence of the hydrogen absorption that occurred by the Sun's ultraviolet light, maximizing at around 90 km (Machol et al., 2019).

To study the MLT summer length, we combine MLT winds from specular meteor radars (SMRs) and mesospheric winds
from partial reflection radars (PRR, Wilhelm et al., 2017; Hoffmann et al., 2010). These systems are located in Germany and northern Norway. Our ground-based observations are complemented with similar years of satellite observations, specifically from the Microwave Limb Sounder (MLS) onboard of the Earth Observing System Aura satellite (e.g. Waters et al., 2006; Wu et al., 2008).

The paper is organized as follows. Section 2 contains the description of the data base and instruments used as well as a brief
description of the applied data processing methods for the individual systems. Section 3 gives a description of the climatology and the applied summer length definitions. The final time series and a brief description of the results are given in Sect. 4. This is followed by the discussion for the individual latitudes and a comparison between the definitions in Sect. 5, and finally, a summary and conclusions are found in Sect. 6.

## 2 Database

Considering the difficulties in obtaining a homogeneous data set, which are well known in long-term studies (e.g. Laštovička and Jelínek, 2019), we used a combination closely located SMRs at two selected latitudes. This combination allows to reduce the number of data gaps and provides a highly reliable wind estimation for the regions under investigation. To study the MZWR at lower altitudes, namely below 80 km, we have included data from PRRs and the microwave limb sounder (MLS) onboard the Aura Earth Observing System (Aura EOS). Both types of radars, SMRs and PRRs, have a good agreement between 80 and
90 km (e.g. Hoffmann et al., 2010; Wilhelm et al., 2017). In the next subsections we briefly describe each of these systems and the methodology of the processed radar data.





## 2.1 Specular meteor radars

SMRs detect meteor trails between mostly 75 and 110 km altitude, measuring their position in space and radial velocity to derive the mean background winds (e.g. Hocking et al., 2001). To generate a homogeneous time series without gaps, we use

a combination of detections from two closely located SMRs, using quasi-simultaneous detections binned in the same way as a single radar mode does to obtain the hourly winds (for details see Chau et al., 2017). This combination helps us to reduce the observation gaps for the high- and mid-latitude regions we have selected. At mid-latitudes (ML) we combine two SMRs located in Germany, specifically Juliusruh (54.6° N, 13.4° E) and Collm (51.3° N, 13.0° E) (e.g. Hoffmann et al., 2010; Jacobi et al., 2015). At high-latitudes (HL) we use two SMRs in northern Norway: Andenes (69.3° N, 16.0° E) and Tromsø (69.6° N,

19.2° E) (e.g. Singer et al., 2004; Hall et al., 2005). The covered years at mid- and high-latitudes are 2005-2020 and 2004-2020, respectively.

Given the inherent variability within the radar measurements the wind dataset of 1 h resolution was first smoothed by a 16-day-width sliding window. The smoothing suppresses short term fluctuations, which are caused by e.g. gravity waves and tides as well as instrumental effects, which are not within the focus of this study. For this long-term study dealing with a length of up

to 31 years, the principal components analysis (PCA, Jolliffe and Jackson, 1993; Jolliffe, 2002) proofed to be a useful tool to compress the data. At each station and for each year, the zonal wind between DOY 100-280 and 82-98 km in the time-altitude depiction is arranged into a 2-dimensional matrix and decomposed as a linear combination of principal components. The first two principal components capture 97.6–99.5% of the total variance and are used to reconstruct the 2-dimensional matrix used for this study and effectively reducing its short-term variability. The principal components representing the dataset are planned

to be investigated in more detail in respect of their temporal evolution in a subsequent study.

## 2.2 Partial reflection radars

PRRs use the mechanism of partial reflection through the ionized component in the atmosphere as a tracer for the neutral motions in the MLT between 50 and 100 km altitude, depending on the instrument configuration and by means of the solar and geomagnetic conditions (see e.g. Fukao and Hamazu, 2014; Reid, 2015). The Saura PRR, located in Andøya (69.1° N,

16.0° E), has been in operation since 2004 (e.g. Singer et al., 2005; Renkwitz and Latteck, 2017). For the mid-latitudes, we use measurements from Juliusruh PRR (54.6° N, 13.4° E) that were obtained between 2004 and 2020 with a comparable system and method (e.g. Hoffmann et al., 2010). To complement our work, we also include data from the Juliusruh PPR predecessor system, using a different configuration and technique between 1990 and 2003. More details on this dataset can be found in Keuer et al. (2007).

Equivalently to the descriptions given for the SMR data, we implemented a 16-day sliding window and the PCA capturing 98.2-99.3% of the total variance with the first two principal components. The time window implemented in the PCA is DOY 50-280, and 70-95 km.



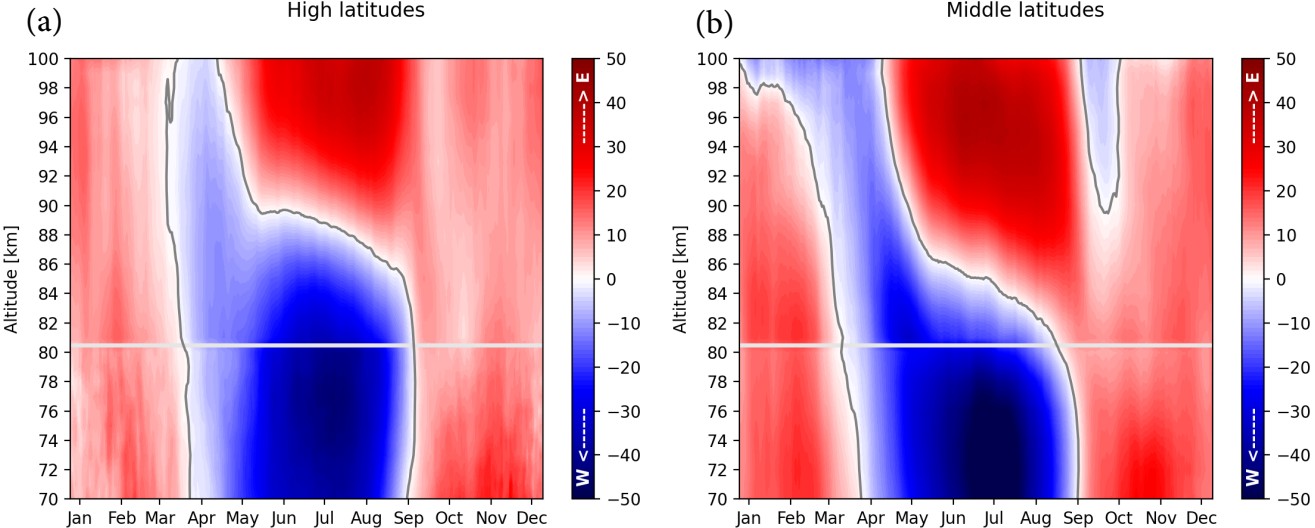

**Figure 1.** Combined mean zonal wind climatologies at (a) high- and (b) mid-latitudes. High-latitudes values above 81-100 km depict the combination of SMRs at Andenes and Tromsø, while below (i.e., 70-80 km) the climatology comes from the Saura PRR. The mid-latitude values are divided in the same altitude range with SMRs at Juliusruh and Collm (81-100 km) and Juliusruh PRR for lower altitudes (70-80 km). The grey line shows the zero wind line.

## 2.3 Microwave limb sounder

Onboard the Aura EOS satellite is the microwave limb sounder (MLS) instrument, sensing atmospheric temperatures from the

troposphere up to 90 km (e.g. Waters et al., 2006; Livesey et al., 2015). From these measurements one can calculate the zonal mean geostrophic zonal winds and geopotential heights (e.g. Yamazaki and Matthias, 2019). In this work, we use MLS zonal mean zonal winds at mid and high-latitudes between 2005 and 2020. It is important to consider that the zonal mean geostrophic zonal wind is a longitude average, while the radars are located at a specific longitude. These time series are extracted with a 16-day sliding window at 74 km (55° N) and at 82 km (70° N).

## 3 MLT mean zonal wind climatologies and summer length definitions

A mean zonal wind climatology for both latitudes and combination of stations is shown in Fig. 1. The high-latitudes climatology (Fig. 1a) is generated from the combination of Andenes and Tromsø SMRs above 81 km and below from Saura PRR. An equivalent approach is used for the mid-latitudes (Fig. 1b), where between 81 and 100 km observations from Juliusruh and Collm SMRs are used and for altitudes below 81km the Juliusruh PRR wind climatology (years 2004-2020). During summer

months, the mean zonal wind over these sites is expected to be equivalent to the zonal mean zonal wind (e.g. Hoffmann et al., 2010).



Both climatologies depict a reversal of the wind (grey line) around March-April, when the wind reverses from eastward (red) to westward (blue) in all altitudes. Between April and May at high altitudes, the wind changes from westward to eastward (grey line) and the temporal evolution of this reversal occurs rapidly from 100 km down to around 90 km (86 km for mid-latitudes). From early June, the mean zonal wind reverses slowly with decreasing altitude until 85 km (78 km at mid-latitudes) until mid-September. Later on, the wind direction reverses rapidly from westward to eastward from these altitudes downwards, indicating the end of the summer in the MLT in mid-September, around one week before the autumnal equinox. The dynamics of the mean zonal wind displays a clear dependence of altitude with respect to latitude (e.g. Laskar et al., 2017; Conte et al., 2018; Wilhelm et al., 2019). Given this latitudinal dependence, we adjust the selected altitudes in the summer length definitions (described below) accordingly.

Figure 2 displays a climatological mean of the MZWR for all data sets. We are indicating the altitudes used in this work for the different summer definitions (see below). At 69° N, the MZWR from the combination of SMRs is depicted in purple and from PRR in orange lines, while for 54° N the combination of SMRs is shown in red and the PRR in green lines. The climatological zonal mean geostrophic zonal wind value is also represented at 82 km for 70° N (solid blue lines) and at 74 km height for 55° N (dash-dotted blue lines).

### 3.1 Mesosphere and Lower Thermosphere - Summer Length

The MLT-SL definition is established by the MZWR from westward to eastward at both the upper and lower altitudes. The altitudes depend primarily on the temporal evolution of the MZWR, and in consequence on the latitude. These altitudes are chosen where the MZWR occurs rapidly and simultaneously for several kilometers. Considering these characteristics, at high-latitudes the MLT-Summer beginning (SB) is chosen at 96 km height and the MLT-Summer end (SE) at 82 km (Fig. 2, purple line). At middle latitude, the MLT-SB altitude is the same as in high-latitudes (i.e., 96 km), but the MLT-SE was chosen at 74 km, using PRR data (see the combination of the red and green lines in Fig. 2). In both definitions, the summer length is the difference between the SE and SB.

### 3.2 Mesosphere - Summer Length

The M-SL is selected at the same altitude varying only by latitude. The summer beginning and summer ending are considered when the final MZWR occurs from east to west, and later from west to east direction, for high-latitudes at 82 km and for mid-latitudes at 74 km (see Fig. 2, orange and green lines, respectively). The same altitudes are taken for MLS (see Fig. 2 blue lines, solid lines at high-latitudes and dash-dotted lines at mid-latitudes). M-SL has been selected to allow a direct comparison between the MLS and radar observations, and other definitions. Moreover, this definition can also be compared with models since most of them are able to reproduce the observed wind field below 90 km during April and May (e.g. Hoffmann et al., 2010; Conte et al., 2018; Pokhotelov et al., 2018).





**Figure 2.** Mean zonal wind reversal comparison and summer length definitions (MLT-SL and M-SL) at high- and mid-latitudes (HL and ML). The mean zonal wind reversal (0 m/s) extracted from the climatologies from: SMRs (purple) and PRR (orange) at 69° N, SMRs (red) and PRR (green) at 54° N. In addition, the specific geopotential heights with the zero zonal mean geostrophic zonal winds values from MLS (blue) are shown at 70° N (solid line) and at 55° N (dash-dotted line). The black arrows indicate the altitude taken for the MZWR used in the individual definitions.



## 4 Results

Here we briefly describe how the radar data have been processed and the results are obtained. We first calculate the daily mean of hourly winds for each altitude and site. Then the mean is smoothed by a 16-day running window, shifted by one day. In order to compress the data and to further reduce its variability, and to be able to focus on the long-term changes, we implemented a PCA (see details in Sect. 2). With the first two components of PCA, we are able to reconstruct the mean zonal winds year by year and considering only the altitudes of interest, we extract the day of the year (DOY) in which the reversal occurs. Following this analysis, we obtain three different time-series for each latitude and definition (SB, SE, and SL=SE-SB). Each time series is treated with a default standard deviation error from the size of the smoothing window, plus an extra consideration for the years where the MZWR is particularly difficult to assess due to an unclear transition. In these cases, the sum of days during the unclear reversal period is divided by two and manually implemented as an error for the particular value.

Figure 3 presents the detected wind reversal dates at high-latitudes, i.e., MLT-SL (right column, purple lines) and M-SL (left column) for the previously introduced altitude definitions. Note that for M-SL, both MLS (blue lines) and PRR (red lines) results are included. The first three panels for both definitions (Figs. 3a, 3b, 3c and Figs. 3e, 3f, 3g) depict the DOY (ordinate) when the MZWR occurs for every year (abscisa) for SB, SE and SL, respectively.

To explore the long-term behavior, we fit a linear function and apply the Student's t-test (with null hypothesis being slope equal to zero) to investigate if there is a significant slope incorporating the standard deviation error propagation (e.g. Santer et al., 2000). The linear regression (solid line in the same color) is only shown for the summer beginning and summer length (first and third panels, respectively), and enclosed in dashed lines (same color) is shown the expected variability. In the case of SE for both definitions, the linear regression is not shown since we were not able to reject the null hypothesis.

The M-SL (Fig. 3g) for the high-latitudes is found to be $170 \pm 11$ days long using PRR measurements ($173 \pm 12$ days for MLS) with a tendency of $0.46 \pm 0.52$ days per year ($1.23 \pm 0.62$ days for MLS). Most of the variability and trend is introduced by the SB (Fig. 3a) occurring at DOY $93 \pm 10$ days for PRR (3 April) and DOY $88 \pm 12$ days for MLS (29 March) with a tendency to start earlier $-0.62 \pm 0.48$ days and $-1.26 \pm 0.62$ days, respectively. For the MLT-SL (Fig. 3c) we found $136 \pm 8$ days, but with no significant trend starting around 7 May (see Tab. 1 for more details).

The last row for both definitions (Figs. 3d, 3h) represents proxies of possible forcing in the MLT region that occurred during the corresponding previous winter (for MSSW and sPJO) or centered in the mean value for the SB (March, April and May) for Lyman$-\alpha$ line (black line), QBO and ENSO. ENSO is represented with the Oceanic Niño Index (ONI), where the values over $1°C$ are considered as El Niño phase (orange) and under $-1°C$ as La Niña (green) (see e.g. Pedatella and Liu, 2012). The QBO is represented with the winds at $10\,hPa$ from Singapore (scaled by 10 m/s). QBO eastward (QBOe) is shown in red and QBO westward (QBOw) is blue. The MSSWs are represented (in purple) as follows: when a displacement of the polar vortex occurred is indicated by a *"D"*, and in case of a split is indicated by the symbol bow tide. In pink are shown the sPJOs.

The mid-latitude results are shown in Fig. 4 in a similar format. The main difference is that the altitude for the summer end in both definitions is 74 km. There, the M-SL (Fig. 4g) is found to be $162 \pm 7$ days long using PRR measurements ($160 \pm 6$ days for MLS) with almost identical tendency of $0.55 \pm 0.3$ days per year for PRR and MLS. Equivalently to the high-latitudes, this



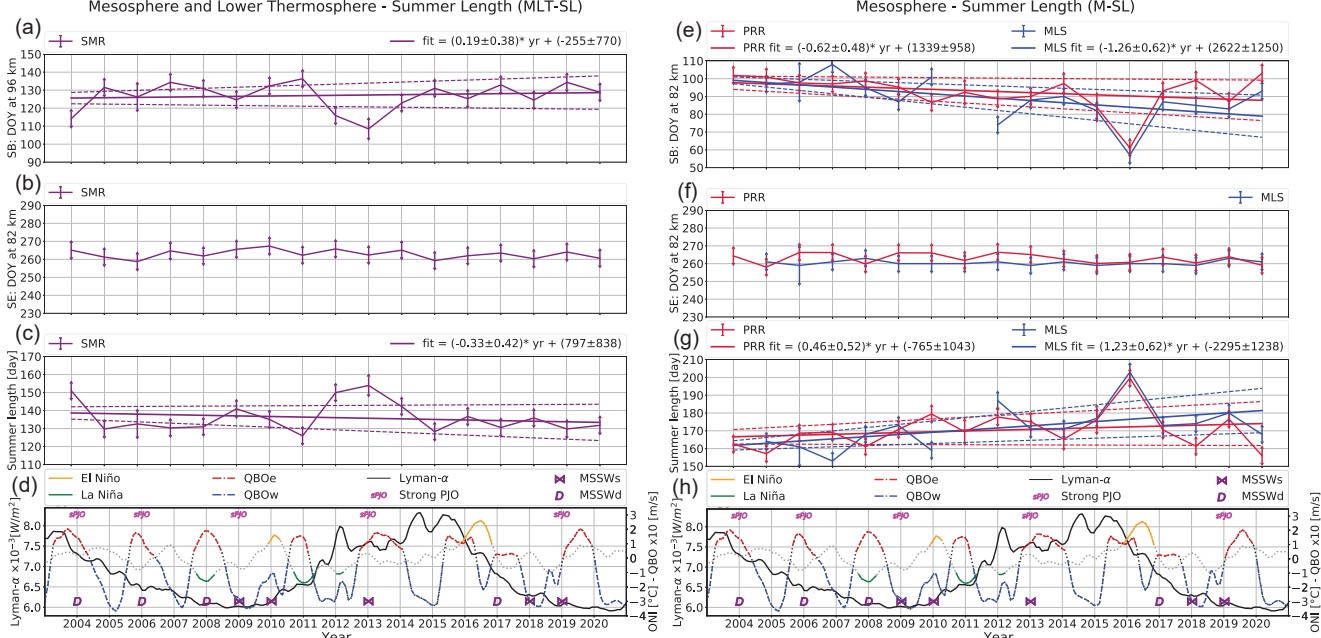

**Figure 3.** Summer length at high-latitudes: On the left is shown MLT-SL and M-SL on the right: (a) represent the beginning of the summer at 96 km (MLT-SB) and (e) 82 km (M-SB) for each year (abscissa) and the DOY when the mean zonal wind reversal occurs (ordinate). The end of the summer is measured at 82 km for both (b) MLT-SE and (f) M-SE. Panels (c) and (g) depicts the difference between SE and SB, i.e., MLT-SL and and M-SL, respectively. For each time series a linear fit is shown in the same color with the standard deviation of the slope in the same color with dashed lines. The panels in the last row, i.e., (d) and (h), show proxies of lower and extraterrestrial atmospheric forcing (see text for details).

mostly corresponds to the SB (Fig. 4a) occurring at DOY $95 \pm 5$ days for PRR and DOY $97 \pm 7$ days for MLS (between 5 and 7 April) with a tendency of $-0.49 \pm 0.25$ days and $-0.72 \pm 0.32$ days, respectively. For the MLT-SL (Fig. 4c) we found $141 \pm 4$ days, but again with no significant trend, starting on 29 April.

The mean values, with the standard deviations from Fig. 3 and 4 are summarized in Table 1, as well as the slopes for the
summer beginning and summer length with their standard deviations. The slope are colored from the result of the Student's-t test, as follows. The slopes with less than 80% of confidence are red, more than 90% is green and greater than 95% are blue.

As mentioned previously, in the case of mid-latitudes one can extend the study of the M-SL using the Juliusruh PPR to 31 years, by combining zonal winds obtained at the same place but with different MF radar systems and measuring techniques. Figure 5 depicts the M-SL values from this combined dataset covering the period 1990-2020. In this case, we show a linear
function for the entire time series in black. Due to an apparent breakpoint around 2008, we include two linear regressions according to this breakpoint, i.e., 1990-2008 in dashed red line and 2008-2020 in the dashed blue line. The breakpoint was obtained by a second degree polynomial fitted by the least square method.

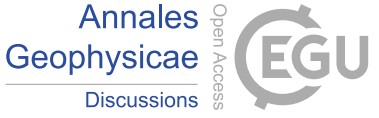

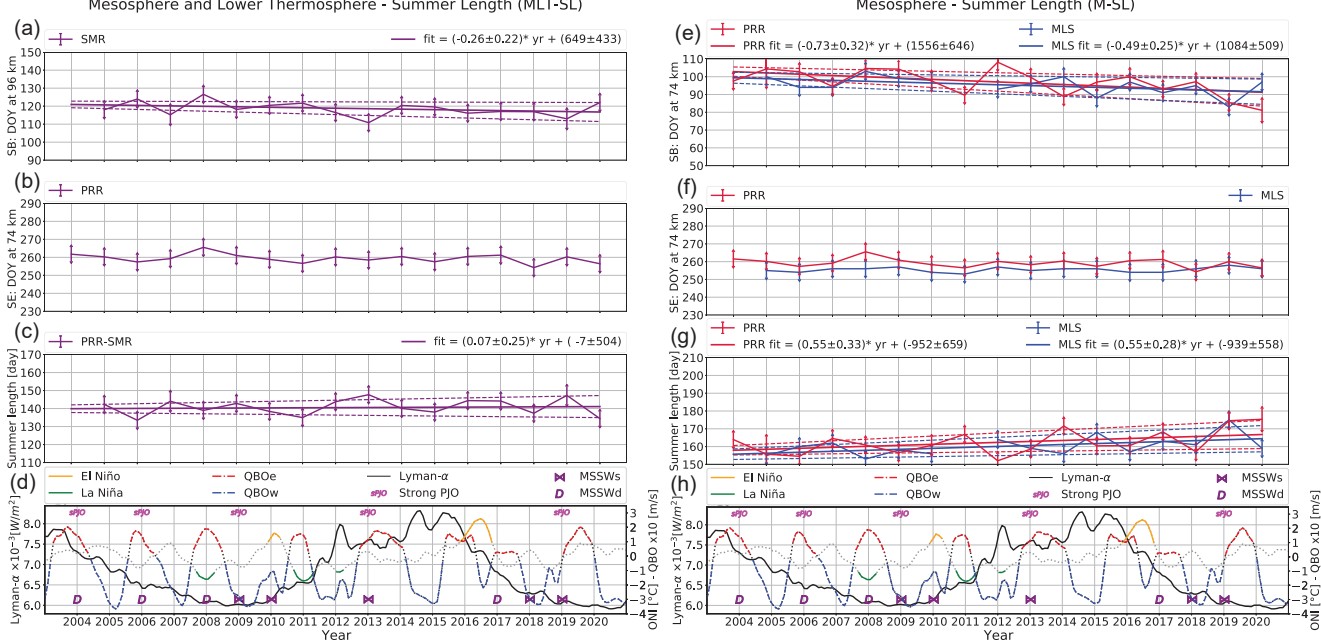

**Figure 4.** Summer length at mid-latitudes: Similar to Fig. 3, but in this case, the altitude of MLT-SE (b) is 74 km, as well as for M-SB (e) and M-SE (g). At this latitude the MLT-SL is obtained from a combination of SMRs and PRR.

## 5 Discussion

In this section we discuss the obtained result. Noteworthy, for both latitudes and definitions, the variability of the summer
lengths are dominated by the summer beginning and thus, by the winter conditions. Since our results display a latitudinal dependency, we also divided our discussion by latitude. In addition, we discuss the results of our summer definitions in respect to other definitions used in earlier studies. The long-term behavior of our results, including the 31-year analysis, is discussed separately.

### 5.1 High-latitudes

As both definitions represent different processes from the different altitudes (in the summer beginning) and therefore times in the year, there is also a significant difference observed in the variability. The observed variability is higher in M-SB due to the proximity to the winter conditions, which is modulated by the planetary wave activity, final warmings, etc. (Lauter and Entzian, 1983; Hoffmann et al., 2002; Savenkova et al., 2012).

The MLT-SL shows peculiar values for the years 2004, 2012 and 2013, with an earlier MLT-SB. The 2013 winter to summer
transition was reported by Fiedler et al. (2015) as an uncommon year with extreme conditions showing lower temperatures

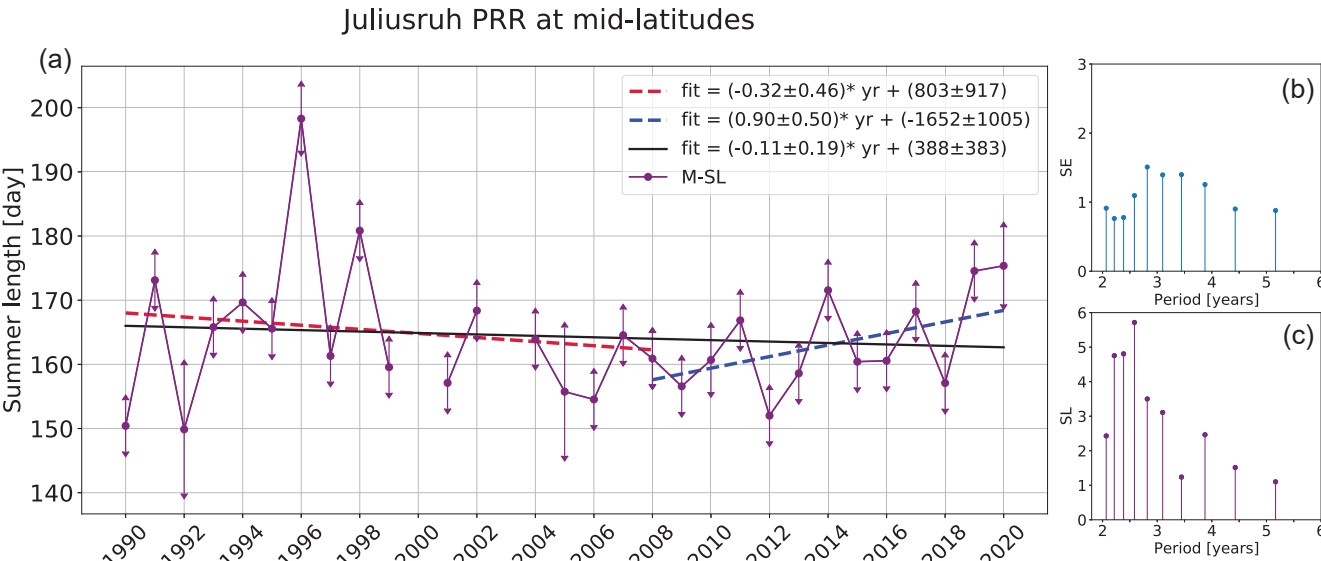

**Figure 5.** (a) Mesosphere - Summer Length at mid-latitudes for 31 years of measurements from Juliusruh PRR. The points show the M-SL obtained from the difference between summer end (SE) and summer beginning (SB) at 74 km. The dashed lines show the linear fit for each interval, from 1990 to 2008 in red and from 2008 to 2020 in blue. The black line denotes the linear fit for the complete time series. On the right, the SE (b) and SL (c) periods in years obtained by the Fourier transformation.

**Table 1.** Summary of the mean values and their standard deviations for each time series with the corresponding values of the slope from the linear fit and confidence values (cv) divided in three categories: less than 80% in red, greater than 90% in green and greater than 95% in blue.

| Def. | Lat. (° N) | Alt. (SB, SE) [km] | System (reference) | SL | SB | SE | SL-Slope | SB-Slope |
|------|-----------|--------------------|--------------------|------|------|------|-----------|-----------|
| MLT | High (69) | (96, 82) | SMR (A-T) | $136 \pm 8$ | $127 \pm 8$ | $263 \pm 3$ | $-0.33 \pm 0.42$ | $0.19 \pm 0.38$ |
|  | Middle (54) | (96, 74) | SMR (J-C), PRR (Jul) | $141 \pm 4$ | $119 \pm 4$ | $259 \pm 3$ | $0.08 \pm 0.25$ | $-0.26 \pm 0.21$ |
| M | High (70) | (82, 82) | MLS (Aura) | $173 \pm 12$ | $88 \pm 12$ | $260 \pm 1$ | $1.23 \pm 0.62$ | $-1.26 \pm 0.62$ |
|  | High (69) | (82, 82) | PRR (Saura) | $170 \pm 11$ | $93 \pm 10$ | $263 \pm 3$ | $0.46 \pm 0.52$ | $-0.62 \pm 0.48$ |
|  | Middle (55) | (74, 74) | MLS (Aura) | $160 \pm 6$ | $95 \pm 5$ | $255 \pm 1$ | $0.55 \pm 0.28$ | $-0.49 \pm 0.25$ |
|  | Middle (54) | (74, 74) | PRR (Jul) | $162 \pm 7$ | $97 \pm 7$ | $259 \pm 3$ | $0.55 \pm 0.33$ | $-0.72 \pm 0.32$ |

Confidence value (cv) > 95%, cv > 90%, cv < 80% .

($\sim 6$K below the mean) and a higher concentration of water vapor at 83 km. In their study, they found strongly enhanced planetary wave activity uncommon for the time of the year (see Fiedler et al., 2015, Fig. 5).

A similar period has been recently studied in the MLT northern hemisphere high-latitude by Hall and Tsutsumi (2020). The authors reported on temperatures at 90 km over Svalbard (78° N) calculating monthly mean values and fitting trends using





December as a representation of winter trend. The authors found lower temperature values in the winters 2003/2004, 2012/2013
and reported 2012 as a change-point detected by their algorithm (see Hall and Tsutsumi, 2020, Figure 3). Even though both
data sets are different (December monthly mean temperature at 78° N and MZWR in May at 69° N), these data "outliers"
appear to be connected.

In the case of M-SL the years 2012 and 2016 strongly deviate from the mean behavior of the M-SL. In 2012 the satellite data

shows an early M-SB (DOY 74) while the radar (DOY 89) depicts a value not far from the mean (DOY 92). At high-latitudes,
in the northern hemisphere the winter is dominated by the behavior of the polar vortex and its temporal dependence of the
position. Considering the satellite zonal mean geostrophic zonal winds are an average in longitude and the radar only shows
the mean zonal wind at a fixed longitude, it is reasonable to find differences between these individual observations. Thus, here
we can see the complexity of understanding the winter time in a localized position (radar site) compared with the average in

longitude (obtained from satellite). In contrast to this example, 2016 has an earlier M-SB for both instruments (DOY 61 PRR
and DOY 57 MLS), as a consequence of the event categorized as a final warming (Yamazaki and Matthias, 2019). Contrary to
this categorization, Manney and Lawrence (2016) described the 2016 as a major final sudden stratospheric warming.

In the time series we were not able to find a relation to Lyman$-\alpha$. In the case of ENSO and QBO, we also do not find a
clear connection with the MZWR dates. A similar result is obtained analyzing the MSSW, sPJO years and the SB time series.

We can only find two particular clear cases where the SB is affected by a final warming.

### 5.2 Mid-latitudes

As we move far away from the polar vortex and approach the mid-latitudes, the summer beginning displays less variability than
at high-latitudes and there is a clear time-latitude difference in the time series (also indicated in Fig. 2). The MZWR occurs
earlier at high-latitudes and later on at mid-latitudes. Towards the end of the summer, the westward wind velocity decreases

and finally reverses again to eastward direction at mid-latitudes and later on at higher latitudes. Thus, the summer length
difference is dependent on the latitude, as a consequence on the residual mesospheric wind circulation (e.g., Andrews et al.,
1987; Hoffmann et al., 2002). The MZWR for both latitudes exhibit a comparable profile, while the MZWR at high latitudes
occurs at about 5 km higher altitudes (see Fig. 1 and Fig. 2). Thus, for the use of the same upper altitude (96 km), the MZWR
occurs earlier in mid-latitudes. Furthermore small differences in the profile steepness are visible, i.e. the wind reversal doesn't

occur simultaneously at several altitudes. However near and especially above 100 km altitude the meteor count rates decrease
substantially, introducing larger uncertainties, which restrain us from selecting a higher altitude (e.g. Younger et al., 2009),
where we exactly want to observe the MZWR and MLT-SB.

Looking into the unusual years seen at high-latitudes, the reversal during 2012 (Fig. 4.a) occurs on the DOY=116, repre-
senting an earlier start but within the variability. However, the reversal occurs in the same day at both latitudes, raising the

question to what kind of event might produce a reversal of the wind in the same day at 15 degrees latitude difference. The
2013 shows a deviation of eighth days apart (DOY 111) from the mean behavior in the MLT-SB, but well outside of the mean
standard deviation of 4 days. This difference shows again evidence of latitudinal difference and the earlier starts, once more
could be representing an unusual strong planetary wave activity for this time of the year (Fiedler et al., 2015). The M-SB (Fig.





4e) depicts a higher variability than MLT-SB, but the only year with a slight deviation from the mean behavior is 2012 with a
late final warming. In addition, this late final warming and the difference between the satellite and the radar could be indicating
the displaced polar vortex near the radar site.

### 5.3 Comparisons to other definitions

Comparing the definitions proposed in this work with the one made by Offermann et al. (2004), we can find a big difference
for the summer end. While Offermann et al. (2010) showed opposite sign slopes retrieved from a threshold in temperature in
the beginning and end, we have found a variability dependence on the summer beginning. This difference is attributed to the
rapid wind changes in September, meanwhile the temperature appears to change with a weaker gradient. The summer duration
obtained in their works is comparable with the values obtained for M-SL at mid-latitudes, with a difference of around ten days.

Inspired by the comparison between the summer duration in the MLT and that at ground level made by Offermann et al.
(2004), we investigate the summer length through the vegetation growing season length. A recent study on the topic has been
performed by Hurdebise et al. (2019) with the tree *Fagus sylvatica* in eastern Belgium between the years 1997 to 2014 obtaining
a leafed length of $165 \pm 7$ days and a slope of $-0.62$ days per year with high significance (statistical p-value $< 0.05$). Chen
et al. (2019) studied three different species of trees (between them the Fagus sylvatica, presenting less change rate between the
studied species) common in central Europe ($47°$ N-$55°$ N), between 1950 and 2013. They found that since 2000 the length of
the growing season has not increased, and their mean value is around $174$ days. These studies show a connection between the
temperatures and the leaf unfolding and folding period at ground level for mid-latitudes. Zhou et al. (2001) extrapolated value
gives us a mean of the vegetation growing season length of $160 \pm 4$ days (maintaining their own standard error), comparable
with M-SL at mid-latitudes ($160$-$162 \pm 7$ days). In the case of Hurdebise et al. (2019) the resulting length of the vegetation
growing season is $169 \pm 7$ days and for Chen et al. (2019) is $174$ with an apparent variability of around 3 days. In these cases,
the M-SL is more similar to the length obtained at high-latitudes ($170$-$173 \pm 12$ days), even though their studies were performed
for mid-latitudes.

### 5.4 Long-term analysis

A linear regression was implemented for all the time series and the result proved with a Student's t-test. Since, in all the
summer ending times series we were not able to reject the null hypothesis, we only show the significance levels for the summer
beginnings and summer lengths. However, the MLT-SL definition shows no significant linear change over the years and thus,
we consider this definition is not sensible to a possible long-term change. On the other hand, the M-SL and M-SB shows linear
tendencies with confidence greater than 95% in most of the cases. The only exception is in M-SL at high-latitudes (Saura
PRR), where the slope shows a confidence greater than 90% (see Table 1). In none of the time series, we applied a correction
by QBO or solar activity as they were used in others works (e.g. Offermann et al., 2010; Keuer et al., 2007). In the case of
QBO, the influence is not clear or seen in the MZWR dates, probably due to the short time series. Pursuing this concept, we
extended M-SL at mid-latitudes with the available data, obtaining a 31-year time series (see Fig. 5a). The summer beginning
revealed periods of $2.21$-$2.58$ years, and the summer end periods of $2.82$-$3.87$ years. Due to the low amplitude in the summer





end, the summer length only reveals those found in the summer beginning (see Figs. 5b and 5c). These periods are associated with QBO and ENSO, 2.2-2.4 yr and 3.5 yr, respectively (e.g. Offermann et al., 2015).

The slope of the 31-year linear regression (negative), shows an opposite direction (positive) than the shorter version (17-year time series, Fig. 4.g) which made us speculate of a non-uniform trend. An inflection point is detected around $2008 \pm 2$ years and the high variability occurs mostly because of the higher uncertainties in the earlier years of the data set, where the radar experienced several changes (see Sect. 2). These uncertainties may influence in the determination of the breakpoint year, and we might be in the presence of another breakpoint around 1992-1995. Evidence of breakpoint in the long-term studies has been reported by several authors. Lauter and Entzian (1983) speculated period of 10-20 years after finding a breakpoint in 1980. The same year was identify by Offermann et al. (2004, 2005) and Offermann et al. (2006) reported an additional one in 2001/2002. Studying the amplitudes of the mean zonal winds during different seasons, Liu et al. (2010) and Jacobi et al. (2015) described a breakpoint in the summer months around 1995-2000 in Collm observations. Later on, Hall and Tsutsumi (2020) detected a breakpoint in $2012 \pm 1$ year. Portnyagin et al. (2006) found a breakpoint in 1980 and adjusted two different linear functions and parabolas concluding that at mid-latitudes, the MLT winds have non-uniform trends. With our initial time series (2004-2020), we can not see a clear indication of a breakpoint. Nevertheless, it is detected within 31 years of measurements. Considering the numerous studies and our findings, we can only assume that a more robust trend analysis might require a longer time series.

## 6 Summary and conclusions

Smoothed mean zonal winds between 2004/2005 and 2020 from different radars located at high- and mid-latitudes (Andenes SMR - Tromsø SMR, Saura PRR, Juliusruh SMR - Collm SMR and Juliusruh PRR) as well as MLS measurements, are used to study two different summer length definitions (see Sec. 3). The MLT-SL definition is taken when the last wind reversal occurred from west to east at 96 km (MLT-SB) and 82 km at high-latitudes (74 km for mid-latitudes) as MLT-SE. On the other hand, the M-SL definition is taken at the same altitude (M-SB and M-SE), but depending on the latitude (82 km at high-latitudes and 74 km at mid-latitudes), when the mean zonal wind reverses from east to west (M-SB) and again, from west to east (M-SE).

With the obtained time series, we analyzed the summer length, studied the variability and the linear tendency. We looked into the dates and the different events occurring in the upper and lower atmosphere, to understand the events modifying the summer length. Furthermore, we compared the summer length to the ground level growing season. The results are summarized as follows:

- The summer length is determined by the MZWR, which depends on the actual latitude and altitude. High-latitudes showed more variability than mid-latitudes for both definitions. The summer beginning presents most of the variability that is transferred to the summer length. The summer end occurs for all latitudes in the same week, before the autumn equinox and presents no significant linear trend.

- MLT-SL definition: the summer starts around 7 May at high-latitudes (SL=136 days) and around 29 April at mid-latitudes (SL=141 days), showing a shorter summer length at high-latitudes. This definition presents no significant trends



and the events studied (MSSW, sPJO, ENSO, QBO and Lyman-$\alpha$), do not seem to affect the duration of the summer.
Nevertheless, we have found strong evidence of abnormal behavior in the years 2004, 2012 and 2013, also observed by
Hall and Tsutsumi (2020). Particularly the year 2013 has been reported by Fiedler et al. (2015) to present high planetary
wave activity later than the usual time, producing an earlier MZWR.

– M-SL definition: is more variable than the MLT-SL due to the higher variability in the summer beginning, which is more
prone to the winter conditions. The summer starts between the end of March and the beginning of April for high-latitudes
and one week later mid-latitudes (see Table 1). In this case, linear trends were found for summer beginning and summer
length with 90% or more confidence. The years 2012 and 2016 displayed extreme values. However, in the latter, the
earlier summer beginning was a consequence of a final warming (Yamazaki and Matthias, 2019).

– At mid-latitudes, the length of the growing season at ground level is similar or has around 10-day difference (depending
on the author) to the M-SL.

– After analyzing the time series and trying to relate it to other events (solar activity, QBO, ENSO, sPJO and MSSW), we
were not able to find a direct influence on the summerl length of summer beginning. Only for the M-SL we found one
year (with a strong MSSW and 2016 final warming) being directly affected. The 17-year time series are short to study the
period related to QBO or ENSO. On the other hand, with the 31-year time series (see Fig. 5), we detected periodicities
of around 2.21-2.58 and 2.82-3.87 years that we could attribute to QBO and ENSO, respectively (e.g. Offermann et al.,
335   2015).

*Data availability.*  The data will be shared by https://www.radar-service.eu/radar/en/home through a DOI.

*Author contributions.*  JJ, TR, JC and PH developed the idea and helped in the interpretation of results. MH assisted in the implementation
and interpretation of PCA. VM and YY provided the wind analysis used for the Microwave Limb Sounder values. CH and MT ensured the
operation of the Tromsø specular meteor radar and CJ of the Collm specular meteor radar. Furthermore, JJ wrote the manuscript with input
from all the coauthors.

*Competing interests.*  CJ is one of the editors-in-chief of Annales Geophyicae. The authors declare that they have no conflict of interest.

*Acknowledgements.*  This work was partly supported by the German Research Foundation (DFG) through grant VACILT (PO 2341/2-1) and
by the Federal Ministry for Education and Research (BMBF) under grants TIMA (01 LG 1902A) in the frame of the Role of the Middle
Atmosphere in Climate (ROMIC)-program. CJ acknowledges support by Deutsche Forschungsgemeinschaft through grant JA 836/47-1
(VACILT).





The QBO winds were obtained from the Free University of Berlin repository (http://www.geo.fu-berlin.de/en/met/ag/ strat/produkte/qbo/). ENSO index (ONI) was acquired from NOAA/National Weather Service (https://origin.cpc.ncep.noaa.gov/products/analysis-monitoring/ ensostuff/ONI-v5.php). Lyman$-\alpha$ values were retrieve from NASA (https://omniweb.gsfc.nasa.gov/form/dx1.html).



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
