# Peer review of "Long-term studies of MLT summer length definitions based on mean zonal wind features observed for more than one solar cycle at mid- and high-latitudes in the northern hemisphere"

_Annales Geophysicae, 2021_

## Author Comment (AC1)

**Response to RC1**
Comment on angeo-2021-51
Anonymous Referee #1
* * *
Referee comment on "Long-term studies of MLT summer length definitions based on mean zonal wind features observed for more than one solar cycle at mid- and high-latitudes in the northern hemisphere" by Juliana Jaen et al., Ann. Geophys. Discuss., https://doi.org/10.5194/angeo-2021-51-RC1, 2021
* * *
This is interesting paper, which is mostly well done, and only minor revision is requested.

Thank you.

Comments:

Figure 3: Right and left columns in figure, its capture and body of text do not agree each other.

This was indeed a mistake, it's corrected now.

Lines 181-187: You describe what is in panels 3d and 3h but no results which follow from these figures are presented.

We actually couldn't find any clear indication of a relationship between the dates of the mean zonal wind reversal and the different events occurring in the atmosphere.
A corresponding statement is between lines 233-235. What was missing and is now added to the manuscript is the equivalent statement for mid-latitudes. Thanks.

Figure 5: Why is break in 2008; visual inspection of Fig. 5 suggests rather 2006. However,I agree with you that it is not possible to determine the break point quite accurately and your interval 2008 ± 2 years includes year 2006.

We detected the year 2008 with a second degree polynomial fitted by the least square method, explained in lines 201-202.

Wording and significant misprints:

Line 107: "Juliusruh PPR" should be "Juliusruh PRR"
Line 266: "leafed length" should be "leafed season length"

Line 331: "summerl length of" should be "summer length or"

Thank you, we corrected these typos.

---

## Author Comment (AC2)

**Response to RC2**

Comment on angeo-2021-51
Anonymous Referee #2

Referee comment on "Long-term studies of MLT summer length definitions based on meanzonal wind features observed for more than one solar cycle at mid- and high-latitudes in the northern hemisphere" by Juliana Jaen et al., Ann. Geophys. Discuss., https://doi.org/10.5194/angeo-2021-51-RC2, 2021

Reviewer Report on the manuscript angeo-2021-51

Long-term studies of MLT summer length definitions based on mean zonal wind featuresobserved for more than one solar cycle at mid- and high-latitudes in the northern hemisphere

by J.Jaen, T Renkwitz, J.L.Chau, et al.

**General Remarks**

- 1) The paper presents an interesting data set of zonal winds obtained from several measurement stations.
- 2) The data are interpreted in terms of the "summer length" and compared to other data.
- 3) Different definitions of "summer length" are used, part of which are not convincing.
- 4) The paper is difficult to read because of the many abbreviations used.
- 5) The paper can be recommended for publication after major changes have been made.

We appreciate the valuable comments given by the reviewer, please see specific comments and answers below. Corresponding changes in the manuscript are highlighted.

**Major Comments**

1) The definition of the summer length in the MLT is dubious. If I understand your text correctly you determine the begin and end of summer at two different altitudes (as well at mid as at high latitudes). A justification is not given. At both times the wind reverses from westward to eastward. This is confusing, as at lower altitudes opposite reversals are used for begin and end of summer. You can, of course, analyze your data this way, however, you should not call this a "summer length". And has it a meaning?

> By the way: The high altitudes in Fig.1 indicate a semi-annual oscillation which is difficult to reconcile with a summer/annual oscillation.

Indeed we propose two different and relative summer length definitions. The concept of a Mesosphere-summer length is equivalent to earlier studies by other authors, e.g. Offermann et al. (2010), investigating a potential long-term change in the summer length. Since a general definition of and MLT region summer does not exist, we propose to relate to specific persistent observed characteristics of atmospheric dynamics, namely the pronounced wind reversals.

We understand the two definitions for the Mesosphere and the mesosphere - lower thermosphere are potentially confusing given they are for different altitudes and have a different direction/orientation and times for the summer beginning. Both altitudes show distinct characteristics, so they were evaluated in detail. As shown in the manuscript, the lower altitude is strongly affected by the winter conditions.

Noteworthy, this "spring" wind reversal in the mesosphere is often well captured by models, which however is not the case for the lower thermosphere (above 90km).

Therefore, we see an important contribution not only to long-term studies but also to the inherent climatology of our study.

2) Fig. 5a: The data after 2008 show a clear four-year signature after the "break". If there is a real break in the atmosphere one might calculate the spectra separately for the two intervals. The four-year oscillation has been analyzed in detail recently by French, Klekociuk, and Mulligan, ACP 2019. The time interval before 2008 has been analyzed for the summer duration by Offermann et al. 2010. These authors find a substantial increase contrary to what is shown in Fig. 5a.

Thanks for the valuable suggestions and comments.

1. Calculating the spectra in the different intervals: Indeed, this can be done, and we did so. The result is quite similar to the one shown in Fig 5b and 5c, periods near 2 years for the first and 3 years for the latter time series are found.

[Figure]

Part of the idea of analyzing the complete time series and exploring the periods was to identify the periods of known oscillations with a robust number of samples including the dates of the reversal.

Regarding quasi-quadrennial oscillation (QQO), we were able to identify a 4.5-year periodicity in the summer beginning 2008-2020 shown on the right, in which the panels depict the summer beginning (SB), summer end (SE) and summer length (SL), respectively. Nevertheless, the amplitude is not as relevant as the 3-year one. We included the comment in the discussion.

We identify a missing constant factor in the code (square root of 2), so we have changed the plots and the amplitude magnitudes are a little bigger now. Thanks for these comments.

2. Time series studied by French, Klelociuk, and Mulligan, 2020 and Offermann et al, 2010:

French, Klelociuk, and Mulligan (2020) studied the correlation between OH temperatures, meridional and zonal winds. They found a positive correlation between temperatures and meridional winds, but for the zonal winds, the correlation was not significant. The citation and comment has been added to the manuscript.

The only time-series that is comparable to ours is the one found in Offermann et al. (2010). They studied the "spring point" where the mean zonal wind reverses at 94 km (at mid-latitudes) between 1992-2008. In this case, it could be comparable to the summer beginning at mid-latitudes for the MLT-summer length (measured at 96 km, 2005-2020), but there are only 4 years of overlap (2005-2008). Both time-series were obtained from two different systems: specular meteor radars, and partial reflection radars (also known as MF). Wilhelm et al. (2017) did a comparison between SMR and PRR in the range of 78-100 km. They found the highest agreement for zonal and meridional winds for these two instruments in the range of 78-94 km, except during the spring-time (or transition) and the summer below 82 km. For this reason, we are not surprised to find differences between the "spring point" in Offermann et al. (2010) and our time series (MLT-SB at mid-latitudes).

Fig.5 b) and c): Please elaborate on these spectra. What are the ordinates? Are there no longer periods?
The ordinates are amplitudes (now added to the manuscript). We can interpret it as magnitude in days. Indeed there are longer periods, but we show the ones that we are referring to. On the right is the complete period for summer beginning, summer end, and summer length, respectively.

[Figure]

3) To read the paper is somewhat strenuous because of the many abbreviations used. I have counted more than a dozen of them. Please try to reduce this, at least in the text.

We understand this comment, we have replaced most of the abbreviations in the text as possible.

4) The title of Section 5.2 "Mid-latitudes" is misleading. The text discusses the MLT as well.

We think that perhaps there was a word confusion. Between lines 205 and 206 is explained that the results are divided by latitude (middle and high latitudes) for both definitions (Mesosphere-SL and Mesosphere and Lower Thermosphere-SL), and thus the division of Mid- and High-latitudes in the text and the titles.

5) Table 1, Column 4:        Please explain "A-T", "J-C" "Jul"

Thanks for pointing this out, it was modified and deleted since it is not necessary for the manuscript and it could potentially cause confusion.

Minor Comments

1) The title of the paper is a bit misleading as the text always presents MLT- and M- results.

   In this case, we considered the Mesosphere as part of the MLT, and thus we don't want to be redundant in the title.

2) Fig.3 and 4: The print in the text of the Figures is too small. Part of it can be read only by means of a magnifying glass.

   We apologize for using small fonts. They have been increased.

3) Lines 167-168: left and right columns are interchanged.

   Yes, thanks, it was changed.

4) Lines 266 and 273: Do the two notions "leafed length" and "growing season" mean the same?

   Indeed. The leafed period length corresponds to the time of the year when the studied particular trees start showing green leaves or sprouts until the point that the leaves fall from the trees, which is equivalent to the growing season.
   We have added a clarification, thanks.

5) Line 280:    "definition is not sensible…" Do you really mean "sensible", or rather "sensitive"?
   Thanks, it was corrected.

6) Line 306 and others: "from west to east" Please change to: "from westward to eastward"

   Corrected.

References:

Offermann, D., Hoffmann, P., Knieling, P., Koppmann, R., Oberheide, J., Steinbrecht,W., 2010. Long-term trends and solar cycle variations of mesospheric tem-perature and dynamics. J. Geophys. Res. 115, D18127.http://dx.doi.org/10.1029/2009JD013363.

French, W. J. R., Klekociuk, A. R., and Mulligan, F. J.: Analysis of 24 years of mesopause region OH rotational temperature observations at Davis, Antarctica – Part 2: Evidence of a quasi-quadrennial oscillation (QQO) in the polar mesosphere, Atmos. Chem. Phys., 20, 8691–8708, https://doi.org/10.5194/acp-20-8691-2020, 2020.

Wilhelm, S., Stober, G., and Chau, J. L.: A comparison of 11-year mesospheric and lower thermospheric winds determined by meteor and MF radar at 69 °N, Ann. Geophys., 35, 893–906, https://doi.org/10.5194/angeo-35-893-2017, 2017.